# Parasitic infection increases risk-taking in a social, intermediate host carnivore

Connor J. Meyer [1,2,3 ✉], Kira A. Cassidy[1,3], Erin E. Stahler [1], Ellen E. Brandell[1], Colby B. Anton[1], Daniel R. Stahler[1] & Douglas W. Smith[1]

*Toxoplasma gondii* is a protozoan parasite capable of infecting any warm-blooded species and can increase risk-taking in intermediate hosts. Despite extensive laboratory research on the effects of *T. gondii* infection on behaviour, little is understood about the effects of toxoplasmosis on wild intermediate host behavior. Yellowstone National Park, Wyoming, USA, has a diverse carnivore community including gray wolves (*Canis lupus*) and cougars (*Puma concolor*), intermediate and definitive hosts of *T. gondii*, respectively. Here, we used 26 years of wolf behavioural, spatial, and serological data to show that wolf territory overlap with areas of high cougar density was an important predictor of infection. In addition, seropositive wolves were more likely to make high-risk decisions such as dispersing and becoming a pack leader, both factors critical to individual fitness and wolf vital rates. Due to the social hierarchy within a wolf pack, we hypothesize that the behavioural effects of toxoplasmosis may create a feedback loop that increases spatial overlap and disease transmission between wolves and cougars. These findings demonstrate that parasites have important implications for intermediate hosts, beyond acute infections, through behavioural impacts. Particularly in a social species, these impacts can surge beyond individuals to affect groups, populations, and even ecosystem processes.

[1] Yellowstone Wolf Project, Yellowstone Center for Resources, P.O. Box 168 Yellowstone National Park, WY 82190, USA. [2] Wildlife Biology Program, Department of Ecosystem and Conservation Sciences, W. A. Franke College of Forestry and Conservation, University of Montana, Missoula, MT 59812, USA. [3] These authors contributed equally: Connor J. Meyer, Kira A. Cassidy. ✉email: Connor.meyer@umontana.edu

*T*oxoplasma gondii is a ubiquitous multihost protozoan parasite capable of infecting any warm-blooded species and requires a felid definitive host to sexually reproduce[1]. Infection spreads either through the ingestion of oocysts shed in the environment by a definitive host (e.g., environmentally mediated transmission via water or vegetation), the ingestion of infected tissue cysts in definitive or intermediate hosts[1,2], or, if the fetus survives infection, vertically through congenital transmission[2].

Once an intermediate host is exposed, the infection spreads from the intestinal lining to form cysts in the brain and muscle tissue and acute toxoplasmosis occurs[2]. If acute infection occurs during pregnancy it can lead to birthing complications, spontaneous abortions, and stillbirths; and in young or immunosuppressed individuals can cause fatal encephalitis[3,4]. Immunocompetent individuals generally exhibit no clinical symptoms but will have a chronic lifetime infection due to the presence of cysts[2]. Experimental studies have shown that chronic infections, even in healthy individuals, can lead to increased dopamine[5,6] and testosterone production[7,8]. These hormone changes can cause increased aggression[9,10] and risk-taking behaviour such as increased hyperactive movement, failure to avoid olfactory predator cues (i.e., seeking out instead of avoiding felid urine), and decreased neophobia[7,11–13].

Considering the effects that *T. gondii* infection can have on intermediate host reproduction and behaviour, *T. gondii's* role in wild ecosystem processes are understudied. One of the few studies focused on infection impacts on behavior in a wild mammal, Gering et al. (2021) found that toxoplasmosis was associated with increased boldness in hyena (*Crocuta crocuta*) cubs and that seropositive hyenas of all ages were more likely to be killed by African lions (*Panthera leo*)[14]. That study demonstrated a mechanistic link between toxoplasmosis and an individual's fitness through behaviour and decision-making.

Gray wolves (*Canis lupus*) in Yellowstone National Park (YNP) have been the subject of extensive research over several decades, primarily focused on predator-prey dynamics, population dynamics, genetics, behaviour, and canine pathogens[15]. YNP is a complex multi-carnivore system, where wolves and a definitive *T. gondii* host, cougars (*Puma concolor*), overlap spatially due to high landscape heterogeneity and prey movements[16]. Thus, similar multispecies *T. gondii* transmission pathways as those found between spotted hyenas and lions could be present between wolves and cougars in North American systems, where wolves that spatially overlap with cougars may have increased *T. gondii* transmission risk via direct or indirect contact with cougars. *T. gondii* has been documented in the YNP gray wolf[17] and we seek to understand *T. gondii's* role in this social, intermediate host carnivore using 26 years of gray wolf serological and observational data.

Our first aim was to determine which demographic and ecological factors affect *T. gondii* infection in wolves in YNP. We tested individual demographic factors, including age, sex, social status at the time of capture, and coat color due to their potential variation in disease susceptibility. Previous research has found the risk of *T. gondii* infection increases with age due to accumulating risk of exposure with time[17,18]. The other three wolf demographic factors were included because of their links to certain hormones, which may influence an animal's susceptibility to infection[19]. Sex hormones play a role in infection risk and, once infected, hormone production may be altered[19]; however, other studies found no link between *T. gondii* seroprevalence and sex[14,17,18]. Due to natural variations in hormone levels (testosterone, progesterone, estrogen, etc.) between the sexes[20], there may be differing risks and subsequent behavioral responses to infection. Previous research has found social status (e.g., pack leaders)[21] and coat

color (gray coat color wolves have higher cortisol levels and increased behavioral aggression)[22] linked to varying hormone levels and immune defense[23,24]. To determine if seroprevalence is affected by the amount of spatial overlap with a *T. gondii* definitive host (i.e., cougars), we included an overlap index for each wolf and areas of high cougar density.

Our second aim was to determine if *T. gondii* infection influences wolf behavior. We identified three wolf behaviours associated with greater risk-taking: (1) dispersing from a pack, (2) achieving dominant social status (referred to as becoming a leader), (3) approaching people or vehicles (referred to as habituation), and two causes of death associated with increased risk:(a) intraspecific mortality (i.e., death by other wolves through interpack fights), or (b) anthropogenic mortality (i.e., death by humans due to decreased proximity to humans or human structures). As behavior can be influenced by many factors, we controlled for certain variables in each of the behavior models: sex can influence behaviors such as dispersal, and age can influence the probability of a certain behavior occurring[25]. Northern YNP has very high wolf density, the roads are open year-round, the elevation is lower and provides winter range for ungulates and opportunities for wolf hunters just outside the park boundary. All these factors may affect wolf behavior as the wolves there may have increased opportunities to disperse, to die, and may be more susceptible to habituation. Therefore, we controlled for YNP system (northern or not) as well. In controlling for these factors that may influence wolf behavior, we aim to isolate the influence of *T. gondii* infection on behavior. We tested if serostatus influenced the odds of a wolf performing these behaviors or dying of one of these causes. We discuss the findings from both of our aims, factors influencing *T. gondii* seroprevalence and determining if toxoplasmosis affects wolf behavior, with respect to interspecific disease dynamics and how behavioural changes can impact gray wolves at multiple scales.

Here we found that *T. gondii* infection in wolves was predicted by pack overlap with a definitive host, cougars, and that wolves seropositive for *T. gondii* changed their behaviour to take greater risks—being more likely to disperse and to become pack leaders than seronegative wolves. Due to a wolf pack's social structure, these behaviour changes may cause a feedback loop that leads to pack-level increases in risk-taking with important implications for further disease transmission, interspecific competition with cougars, and wolf survival.

## Results

**Serology.** Of the 62 cougars tested for *T. gondii*, 51.6% were seropositive. Seroprevalence in cougars increased from 45% during the first sampling time ($n = 47$, 1999 to 2004) to 73% during the second sampling time ($n = 15$, 2016 to 2020). This test confirmed the presence of *T. gondii* in YNP's most-abundant definitive host.

Between 1995 and 2020, an average of 11.8 sera samples were collected each year (range = 4–22) to test for *T. gondii* antibodies. All 50 tests from 1995 through 1999 were negative, then three wolves tested seropositive in 2000. Thereafter, between one and eight wolves were seropositive each year. Seventeen equivocal samples were detected using the ELISA and were then rerun using the MAT assay, which allowed us to distinguish eleven seropositive and six seronegative samples. The pooled seroprevalence was 0.0% from 1995 to 2000, 24.5% from 2000 to 2004, 18.7% from 2005 to 2009, 42.9% from 2010 to 2014, and 36.5% from 2015 to 2020. Using samples collected from 2000 to 2020, we ran 273 tests on 256 samples. Prevalence was 27.1% ($n = 74$) with 61.9% negative ($n = 169$) and 11.0% equivocal ($n = 30$). Twenty-five individuals were tested more than once throughout

their life due to multiple captures, and therefore have multiple samples, spaced at least eleven months apart. Eight males were tested twice, 15 females were tested twice, and two females were tested three times. Accounting for multiple tests, 229 individuals were tested: 116 males, 112 females, and one hermaphrodite. Females (31.25%) had slightly higher seroprevalence than males (25.00%), but these proportions were not different (z-score = −1.05, $p = 0.15$).

Wolf age was recorded both as a continuous variable and categorical variable with 100 pups, 53 yearlings, 88 adults (aged 2.0–5.9), and 15 old adults (aged 6.0 and older). Seroprevalence was similar between the three younger categories (pup = 29.00%, yearling = 28.30%, adult=26.14%) and only increased with old adults (46.67%). A test comparing pup seroprevalence to all other age categories was not significant (z-score=0.346, $p = 0.36$). The biggest difference was between old adults and all other age categories pooled (z-score = −1.40, $p = 0.08$). We also tested for differences in *T. gondii* exposure between gray and black coat colors and found no difference: gray wolves ($n = 115$) had 25.22% and black wolves ($n = 114$) had 31.58% seropositivity (z-score = −1.07, $p = 0.14$). Similarly, we tested social status at the time of sampling and found no difference: subordinates ($n = 197$) had 30.45% and leaders ($n = 59$) had 23.73% seropositivity.

Wolves that had at least 42.1% overlap with cougar density ≥1.8/100 km² (HCO) had a higher proportion of seropositive tests than wolves with MCO (5.1–42.0%), which was higher than wolves with LCO (0 to 5% overlap). Twelve wolves ($n = 83$, 14.46%) with LCO were positive, 30 wolves ($n = 83$, 36.14%) with MCO were positive, and 31 wolves ($n = 84$, 36.90%) with HCO were positive. The proportion of seropositive wolves in a pooled MCO and HCO was greater than wolves with LCO (z-score = −3.65, $p = $ <0.001). To visualize cougar density and overlap with different wolf pack territories we pooled seropositive tests in nine general wolf use areas and plotted them on a map of YNP with high cougar density highlighted (Fig. 1).

**Demography analysis results**. The full model testing the probability of seropositivity, with a $w_i = 0.99$, included SEX, AGE IN YEARS, SOCIAL STATUS, COAT COLOR, and COUGAR OVERLAP INDEX (Table 1). The NULL model performed poorly, with a $w_i = 0.01$.

The COUGAR OVERLAP index (β = 1.089, 95% CIs: 0.176–2.003) was an important factor in the odds a wolf was seropositive for *T. gondii*. An increase from LCO to MCO to HCO was associated with a higher likelihood of testing positive. The odds ratio of COUGAR OVERLAP was 2.97 (exp[1.089]), meaning the odds an MCO wolf was seropositive was nearly three times higher than an LCO wolf. The odds an HCO wolf was seropositive was almost 9 times higher odds of being seropositive than a wolf in LCO. Predicted probabilities for seropositivity (Fig. 2), based on the full model, showed that seropositivity increased non-linearly with cougar overlap: wolves living in areas with LCO had a predicted 4.7% prevalence, whereas wolves living in MCO had a predicted 12.5% prevalence, and wolves in HCO had a predicted 28.4% prevalence.

Unexpectedly, AGE did not have an effect on *T. gondii* infection (β = 0.296, 95% CIs: −0.158–0.751) and the 95% confidence intervals overlapped zero. The full model included SEX, but this variable was nonsignificant and the confidence interval overlapped zero (β = 0.769, 95% CIs:−0.4984–2.022). SOCIAL STATUS at the time of testing was non-significant (β = −0.836, 95% CIs:−1.982–0.311) as was COAT COLOR (β = 0.516, 95% CIs:−0.726–1.757).

**Behaviour analysis results**. Wolves classified as dispersers had nearly double the *T. gondii* seroprevalence of non-dispersers: 36.26% for dispersers and 18.42% for non-dispersers (z-score=3.11 $p < 0.001$). The model (DISP₁) that included TOXO performed better ($w_i = 0.92$; Table 2) than the model without TOXO (DISP₂). All four variables were significant with $p$ values < 0.05 and none of the confidence intervals overlapped zero. Males were more likely to disperse than females, wolves living in northern YNP were more likely to disperse than wolves in the interior of YNP, wolves were more likely to disperse with increasing time monitored, and seropositive wolves were more likely to disperse than seronegative wolves (β = 2.459, 95% CIs: 0.298–4.620). The odds ratio for TOXO was 11.69 (exp[2.459]), meaning the odds a seropositive wolf disperses was 11 times higher than the odds a seronegative wolf disperses.

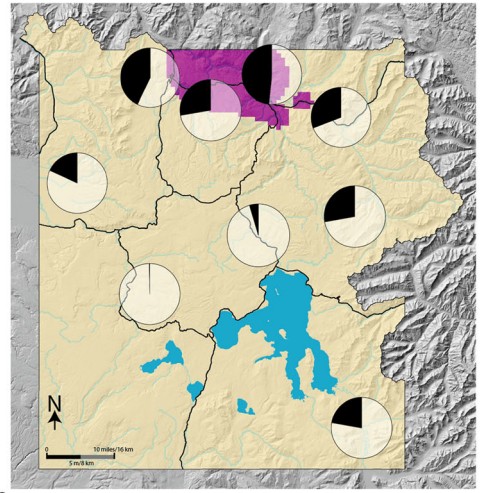
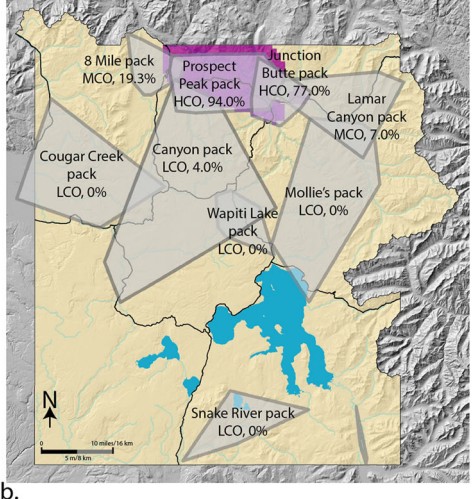

**Fig. 1 Map of cougar density and *T. gondii* seroprevalence in wolves in Yellowstone National Park (YNP). a** Map of cougar density and *T. gondii* seroprevalence in wolves in Yellowstone National Park (YNP). Yellow indicates cougar density <1.8/100 km² and purple indicates cougar density ≥1.8/100 km². Pie charts show the *T. gondii* seroprevalence (seropositive=black; seronegative=white/transparent) from wolves living in nine general areas throughout YNP, pooled across years 2000–2020. **b** A sample year (2015) of wolf pack territory minimum convex polygons in YNP along with each pack's cougar overlap index level (LCO, MCO, or HCO) based on percentage of overlap with cougar density ≥1.8/100 km² (purple).

**Table 1 Full GLMM for the probability that an individual wolf tests positive for *T. gondii*.**

| Parameter | β | SE | z | P | 95% confidence interval for β | |
|---|---|---|---|---|---|---|
| SEX | 0.769 | 0.639 | 1.20 | 0.229 | −0.484 | 2.022 |
| AGE (in years) | 0.296 | 0.232 | 1.28 | 0.201 | −0.158 | 0.751 |
| SOCIAL STATUS | −0.836 | 0.585 | −1.43 | 0.153 | −1.982 | 0.311 |
| COAT COLOR | 0.516 | 0.633 | 0.81 | 0.415 | −0.726 | 1.757 |
| COUGAR OVERLAP | 1.089 | 0.466 | 2.34 | 0.019 | 0.176 | 2.003 |
| intercept | −4.371 | 1.624 | −2.69 | 0.007 | −7.553 | −1.189 |

The reference sex is male, social status is subordinate and coat color is gray. Individual wolf identification number was included as a random intercept.

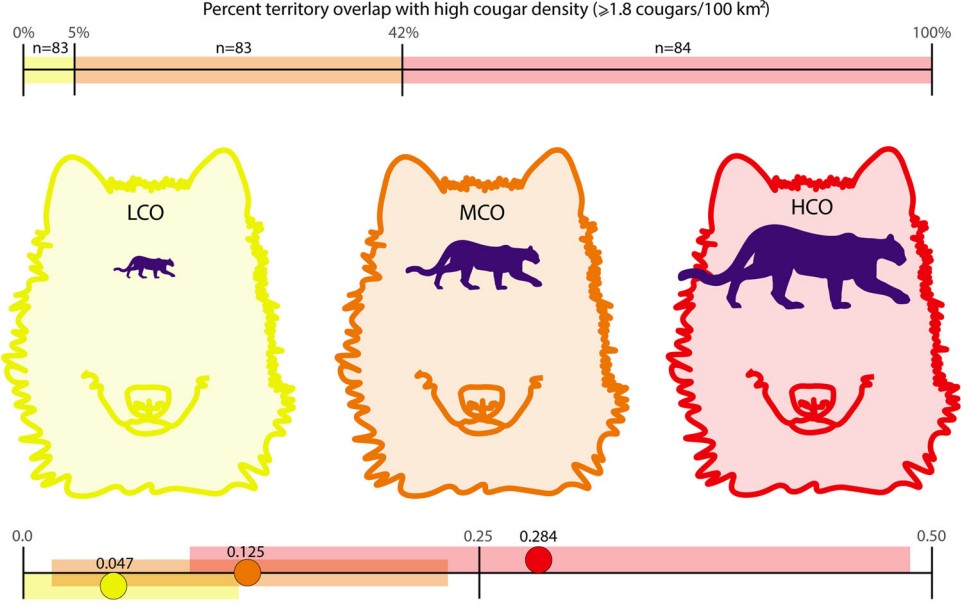

**Fig. 2 Gray wolf *T. gondii* serostatus results and predicted probability of infection given cougar density overlap.** Gray wolves with *T. gondii* serology results were divided into one of three categories relative to their average annual overlap with cougar density ≥1.8/100 km² (top bar): Low Cougar Overlap (LCO in yellow) indicates wolves living in areas with 0.0 to 5.0% overlap, Moderate Cougar Overlap (MCO in orange) indicates 5.1–42.0% overlap, and High Cougar Overlap (HCO in red) indicates 42.1–100% overlap. This results in three categories of nearly equal sample size. The lower bars show the predicted probabilities, with 95% confidence intervals, of a seropositive *T. gondii* test for gray wolves living in LCO (yellow), MCO (orange), or HCO (red). Predicted probabilities are based on the full model.

**Table 2 Best-fit GLMM for the probability that a wolf disperses.**

| Parameter | β | SE | z | P | 95% confidence interval for β | |
|---|---|---|---|---|---|---|
| SEX | −5.251 | 1.837 | −2.86 | 0.004 | −8.852 | −1.649 |
| SYSTEM | −4.590 | 1.744 | −2.63 | 0.008 | 8.007 | −1.173 |
| TIME AVAILABLE | 0.005 | 0.002 | 3.10 | 0.002 | 0.002 | 0.008 |
| TOXO | 2.459 | 1.103 | 2.23 | 0.026 | 0.298 | 4.620 |
| intercept | −2.058 | 0.947 | −2.17 | 0.030 | −3.914 | −0.202 |

The reference sex is male, reference system is northern YNP, and reference *T. gondii* status is negative. Individual wolf identification number was included as a random intercept.

Using the best-performing model to predict dispersal, we found that seropositive male wolves were most likely to disperse, followed by seronegative male wolves, seropositive females, then seronegative females (Fig. 3). This result confirms previously reported evidence of sex-biased dispersal in YNP wolves[25] and indicates that *T. gondii* infection influences the decision to disperse in both sexes. A seropositive male has a 50% probability of dispersing by six months monitored and seronegative males by 21 months monitored. Seropositive females have a 25% probability of dispersing by 30 months monitored whereas seronegative females reach the same probability at 48 months. These differences (15 months for males to reach 50% dispersing and 18 months for females to reach 25% dispersing) suggest seropositive wolves disperse at much higher rates than seronegative wolves.

Seropositive wolves were more likely to become pack leaders than seronegative wolves (z = 4.1705, p < 0.001). The model including TOXO (LEAD₁) performed better (wᵢ = 1.0, Table 3) than the model without TOXO (LEAD₂) and SYSTEM was the only variable that overlapped zero. The effect of TOXO was positive and the confidence intervals just reached zero (β = 3.83, 95% CIs: −0.004–7.664). With an odds ratio of 46.06 (exp[3.83]), the odds that a seropositive wolf becomes a pack leader is more than 46 times higher than a seronegative wolf becoming a pack leader.

Using the best-performing model to predict leadership we plotted the probability of becoming a leader for wolves with and

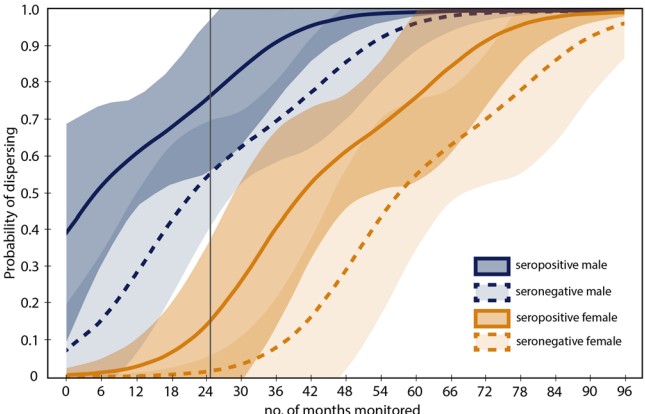

**Fig. 3 Predicted probabilities of wolf dispersal given sex, amount of time monitored, and *T. gondii* serostatus.** Predicted probabilities of dispersing for male (blue) and female (orange) gray wolves with a seropositive (solid lines) or seronegative (dashed lines) *T. gondii* test. The shaded areas indicate 95% confidence intervals. The gray line indicates the average number of months (24.9) a wolf is monitored in YNP.

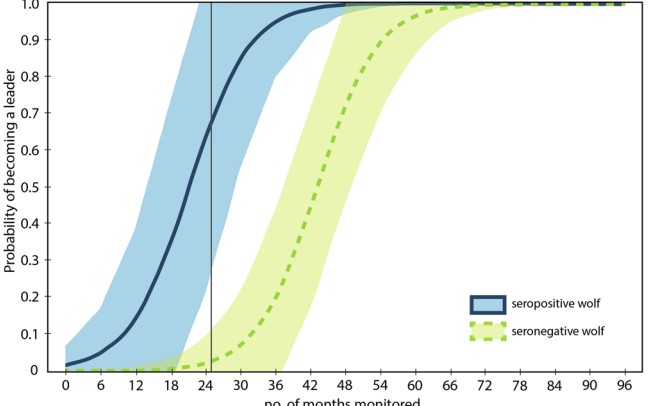

**Fig. 4 Predicted probabilities of wolf leadership given the amount of time monitored and *T. gondii* serostatus.** Predicted probabilities of becoming a pack leader for gray wolves with a seropositive (solid blue line) or seronegative (dashed lime line) *T. gondii* test. The shaded areas indicate 95% confidence intervals. The gray line indicates the average number of months (24.9) a wolf is monitored in YNP.

---

**Table 3 Best-fit GLMM for the probability that a wolf becomes a pack leader.**

| Parameter | β | SE | z | P | 95% confidence interval for β | |
|---|---|---|---|---|---|---|
| TIME AVAILABLE | 0.006 | 0.002 | 2.59 | 0.009 | 0.002 | 0.011 |
| SYSTEM | −1.453 | 1.056 | −1.38 | 0.169 | −3.522 | 0.617 |
| TOXO | 3.83 | 1.956 | 1.96 | 0.05 | −0.004 | 7.664 |
| intercept | −7.696 | 3.07 | −2.51 | 0.012 | −13.713 | −1.68 |

The reference system is northern YNP, and reference *T. gondii* status is negative. Individual wolf identification number was included as a random intercept.

---

without *T. gondii* and found that seropositive wolves were more likely to become a leader than seronegative wolves and this effect increased with time monitored (Fig. 4).

*T. gondii* infection did not explain habituated behaviour ($z = 1.604$, $p = 0.055$) although this sample size was very small with only 27 wolves classified as habituated and only four of those seropositive (13.79%). The model without TOXO (HAB$_2$; $w_i = 0.69$) performed better than the model with TOXO (HAB$_1$). This behavior measure was very course and limited by sample size.

We tested for differences in *T. gondii* seroprevalence for wolves with different cause-specific mortality and found no differences with several comparisons of the proportions: human-caused ($n = 56$, 32.14% *T. gondii* positive) versus natural-caused ($n = 99$; 31.31%) was not different ($z = 0.107$, $p = 0.456$). Wolf-caused ($n = 63$, 31.75%) versus any other known cause ($n = 137$; 33.78%) was not different ($z = 0.253$, $p = 0.401$). And wolf-caused versus only known (non-wolf) natural mortalities ($n = 81$; 38.90%) was not different ($z = −0.567$, $p = 0.284$). For both causes of death examined through GLMMs, the models without TOXO performed better (INTRA-MORT$_2$: $w_i = 0.72$ and ANTHRO-MORT$_2$: $w_i = 0.75$) than models including TOXO (INTRA-MORT$_1$ & ANTHRO-MORT$_1$).

## Discussion

This study provides insights into the relationship between parasite infection and intermediate host behaviour in a wild system. Infection with the parasite *Toxoplasma gondii* has been linked to

increased risk-taking in rodents[7,11,12], chimpanzees (*Pan troglodytes*)[13], hyenas[14], and now gray wolves. Wolf territory overlap with cougars was a major ecological predictor of *T. gondii* infection, while wolf demographics (e.g. sex or age) were not informative. *T. gondii* seropositivity impacted wolf behaviour in two of the three measures of risky behaviour that we tested: seropositive wolves were more likely to disperse and to become pack leaders than seronegative wolves (Fig. 5). These results support not only the historical laboratory work, but also the recent work on hyenas[14] confirming toxoplasmosis can affect behaviour and decision-making in wild intermediate host species[5,11,14].

Gray wolves and cougars are competitors that evolved concurrently in North America[16]. These two carnivores generally prey on the same species, yet interspecific conflict is mitigated through partitioned time and space use, as well as active avoidance of each other[16,26]. Our study corroborates Gering et al. (2021) findings that parasite transmission is an important factor in the dynamic between sympatric carnivores[14].

Due to the strong impact of overlap with cougars on gray wolf *T. gondii* seroprevalence, gray wolves are likely contracting the parasite through direct contact with infected cougars or their shed oocysts, and not through the consumption of alternative intermediate host species. Large ungulates in YNP migrate seasonally to all areas of the park, overlapping all the sampled wolf pack territories. If these ungulates had high *T. gondii* seroprevalence, we would likely record all wolf packs having similar seroprevalence levels. Elk, the primary prey of wolves in YNP, have been tested for *T. gondii* ($n = 155$) and none were clearly seropositive and only five (3.2%) were suspected seropositive (unpublished data, Yellowstone Center for Resources Wildlife Health).

During winter, migrating ungulates in YNP move to lower elevations[26–28] and, as both wolves and cougars hunt ungulates, this subsequently increases overlap between the two predators[16]. This increase in overlap occurs during the wolf breeding, gestation, and whelping seasons (February to June[20]) from mid-winter into early spring. Assuming wolves are more likely to be exposed to *T. gondii* during this time, there is potential for negative impacts on their reproduction. Specifically, acute *T. gondii* can cause complications in embryonic and fetal development and result in fetal or newborn mortality, as shown in domestic dogs[4].

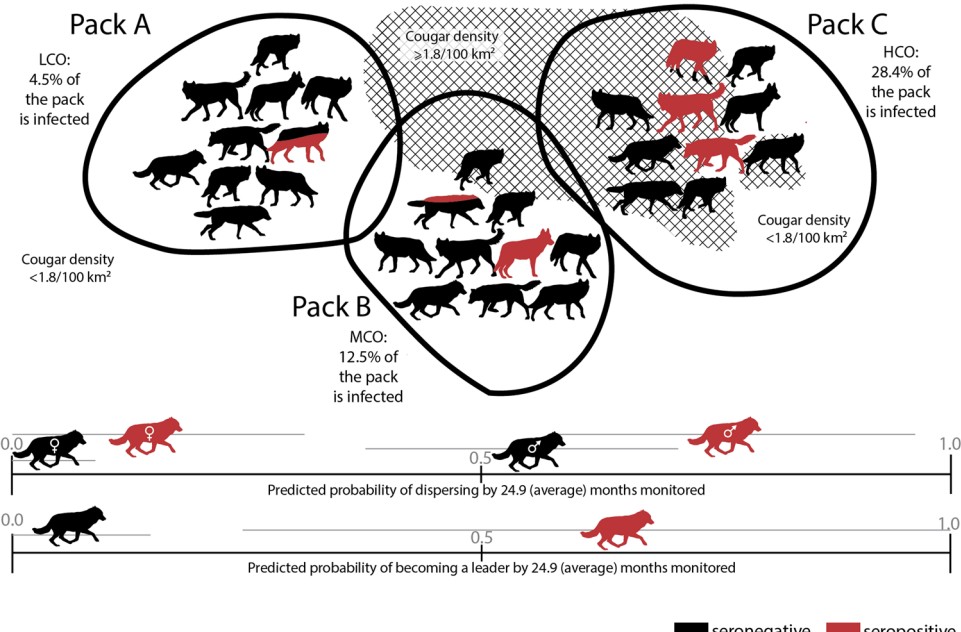

**Fig. 5 Visual depiction of predicted probabilities of wolf serostatus and behavior.** Schematic of results from both the demographic and the behaviour analysis. Displayed at the top are three sample packs with different cougar overlap categories and their corresponding predicted probabilities of *T. gondii* infection (seronegative in black; seropositive in red) based on the best-fit demographic model. Red-filled wolves indicate the expected percent of infected wolves out of 100% (e.g., total number of wolves in the pack). Cougar density ≥1.8/100 km² is depicted with hatch-marks. Cougar density below 1.8/100 km² is all the area outside of the hatch-marks. At the bottom are the predicted probabilities with 95% confidence intervals (gray lines) based on the best-fit behaviour models, of two risky behaviours: dispersing and becoming a pack leader for seronegative and seropositive wolves at 24.9 months monitored (the average number of months wolves in this study were monitored).

In addition, wolf-to-wolf transmission might be possible during this time (the winter wolf breeding season) because *T. gondii* can be sexually transmitted in canid species during acute infection[4].

In congruence with previous *T. gondii* wildlife studies, including wolves, sex was not an important predictor of *T. gondii* infection[14,17,18]. Contrary to other studies[14,17], model results showed that wolf infection did not vary with age. These effects could possibly be masked by spatial differences in exposure whereby a young wolf living in areas with many cougars has a higher risk of infection than an older wolf living for years in a territory with little cougar overlap.

This study is the first to examine the relationship between *T. gondii* infection and wolf behaviour and decision-making, finding a link between parasitic infection and wolf ecology. Dispersal is an important function of wolf population dynamics, but represents a risky decision as those that disperse suffer higher mortality rates[25,29,30]. However, if an individual survives the dispersal process, they often find increased opportunities for reproduction[31]. As seropositive wolves are more likely to disperse, this presents a potential for wolves with *T. gondii* to fill gaps in unoccupied territories or attempt to establish populations in new areas.

Additionally, seropositive wolves were almost twice as likely to become pack leaders compared to seronegative wolves. Having *T. gondii* may increase testosterone levels[7,8] leading to heightened aggression[9,10] and preferential sexual selection (as was found in rats [*Rattus norvegicus*])[7,32]. If seropositive wolves are more aggressive during intra-pack interactions, they may more easily become pack leaders. Increased aggression and dominance, combined with possible preferential sexual selection, may explain the mechanism between toxoplasmosis and leadership. Ultimately, obtaining a dominant leadership position is critical for increased fitness through reproductive success[33,34] and is presumably under strong selection.

Due to the group-living structure of the gray wolf pack, the pack leaders have a disproportionate influence on their pack mates and on group decisions (Fig. 6). If the lead wolves are infected with *T. gondii* and show behavioural changes (e.g., seeking out felid scent or novel situations, as in rats or chimpanzees)[12,13] this may create a dynamic whereby behaviour, triggered by the parasite in one wolf, influences the rest of the wolves in the pack. If pack leaders seek out felid scent, this could increase the likelihood that uninfected individuals encounter infected cougars or their shed *T. gondii* oocysts in the environment and then become infected as well. An intermediate host seeking out felid scent helps enable the parasite to complete its lifecycle if the host was killed and consumed by the felid[12,13]. Additionally, through social learning, pack leader behaviour (i.e., seeking out riskier situations) may be observed and emulated by uninfected individuals, creating a more assertive, risk-embracing pack culture even though only a few key individuals are actually infected. Both pathways could increase wolf *T. gondii* infections through increases in spatial overlap and perhaps even direct interactions with cougars.

There are almost certainly evolutionary limits in place to moderate this feedback loop. Acute infection during pregnancy can result in litter mortality[4]. This aspect of acute infection would severely decrease reproductive success and be evolutionarily disadvantageous for infected individuals. Furthermore, it is rare that a wolf dies of a cause that has little or no risk. The three leading causes of death for wolves in YNP are intraspecific fights, anthropogenic (e.g., hunted by humans, hit by vehicle), and fatal injuries incurred while hunting large prey[25]. If infected wolves (or those that have learned from infected leaders) take greater risks, it is likely their survival will be lower than those avoiding risks. This hypothetical *T. gondii*-cougar-wolf-behaviour feedback loop depends on the balance between risks resulting in an evolutionary advantage (e.g., leadership and

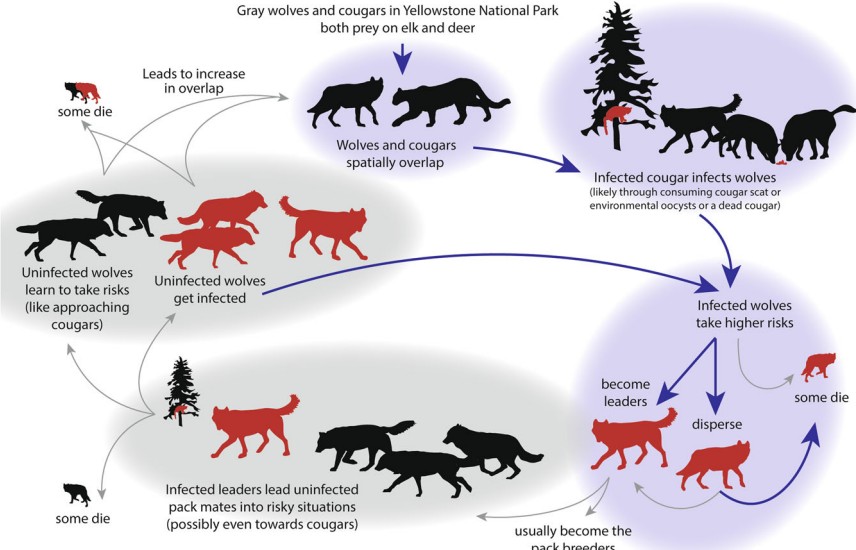

**Fig. 6 Hypothesized wolf-cougar-*T. gondii* feedback loop.** Schematic of the possible feedback loop involving gray wolves, cougars, and *T. gondii*. Red figures indicate seropositive animals and black indicates seronegative animals. Thick, purple arrows indicate links supported by this or other published literature. Thin, gray lines indicate hypothesized relationships.

increased reproductive opportunities) and risks resulting in premature death. Regardless, the effects of *T. gondii* infection clearly ripple beyond our findings in this study with implications for gray wolf survival and reproduction, and interspecific competition and disease dynamics.

This study is a rare demonstration of a parasite infection influencing behaviour in a wild mammal population. We identified a substantial increase in the odds of dispersal and of becoming a pack leader, both risky behaviours, in wolves seropositive for *Toxoplasma gondii*. These two life history behaviours represent some of the most important decisions a wolf can make in its lifetime and may have dramatic impacts on gray wolf fitness, distribution, and vital rates. Dispersing wolves often explore new habitats and are the individuals expanding current gray wolf range. Dispersers that survive to establish a territory often gain breeding positions. Pack leaders are the individuals most likely to reproduce which has important implications on population growth rate and may also affect pack behaviour and culture. Additionally, our results suggest that wolves contract *T. gondii* directly from cougars or their shed oocysts, not through an intermediate host. This study demonstrates how community-level interactions can affect individual behaviour and could potentially scale up to group-level decision-making, population biology, and community ecology. Incorporating the implications of parasite infections into future wildlife research is vital to understanding the impacts of parasites on individuals, groups, populations, and ecosystem processes.

## Method

**Ethics statement**. All wildlife were handled in accordance with recommendations from the American Society of Mammalogists[35]. Wolf capture protocols are approved by National Park Service veterinarians and University of Montana IACUC #AUP 046-21. Cougar capture protocols are approved by National Park Service veterinarians under IACUC numbers IMR_YELL_Stahler_cougar_2018.A1 and 1988-YCW-502.

**Study area**. YNP is a National Park managed by the United States Department of the Interior. It is visited by over 4 million people each year, and over 98% of the park's area (8,991 km²) is undeveloped and managed as wilderness. Northern YNP (995-km²) differs from the interior of YNP (see Houston 1978 for detailed description and map with boundaries)[27] as it has lower elevation and is used by elk (*Cervus canadensis*), bison (*Bison bison*), and mule deer (*Odocoileus hemionus*) as

primary winter range. Consequently, this area has relatively high wolf and cougar densities. The wolf population, after being extirpated in the early 1900s and then reintroduced in 1995 to 1997, generally numbered between 90 to 120 wolves in eight to twelve packs[25], with approximately half the total in northern YNP. Three different felid species that serve as definitive hosts to *T. gondii* have been found in YNP. Cougars are the focus of this study because bobcats (*Lynx rufus*) occur at very low densities (pers. communication K. Gunther) and Canada lynx (*Lynx canadensis*) are extremely rare[36]. Cougars were largely eradicated from the area by the 1930s, but naturally recolonized YNP by the 1980s, creating a resident population primarily in northern YNP[37].

**Sample collection**. Since wolf reintroduction to YNP in 1995, biologists captured and radio collared 12–20 wolves each year to understand wolf movement and ecology. The wolves were anesthetized with a 10 mg/kg dose of Telazol® (tiletamine & zolazepam), categorized as male or female based on external reproductive organs, then fitted with a VHF (Very High Frequency) or GPS (Global Positioning System) radio-collar (Telonics inc. Mesa, AZ; Vectronic Aerospace Berlin, Germany). Approximately 8–30 ml of blood was drawn from the cephalic vein into vacutainer tubes for genetic sampling and disease screening. Between 3–15 ml of the blood was centrifuged for 15 minutes and the serum was extracted and stored at −80 °C in 1.8 ml plastic cryovials. Age estimates were based on tooth wear, pelage fading, and known pack composition. Coat color was recorded as either gray or black.

**Serological screening**. Sera from wolves were tested for antibodies to *Toxoplasma gondii* by the Cornell Animal Health Diagnostic Center (Ithaca, NY). Samples were tested using either enzyme-linked immunosorbent assay (ELISA) or modified-agglutination tests (MAT). For ELISA tests, any optical density measure less than 0.90 was considered negative, 0.91–1.09 was equivocal, and greater than 1.10 was considered positive; for MAT tests, any result with a detectable antibody titer level of 1:25 or greater dilution was considered positive. To be conservative, we treated equivocal tests as negative for analysis. We report prevalence as the percent of seropositive wolves out of the total number of wolves tested. As the first *T. gondii* infection in a gray wolf in YNP was confirmed in 2000, five years after testing began, the data used in the analyses were from 2000 and after.

To confirm that *T. gondii* was present in the cougar population, sera from 62 individuals captured (methods described in Ruth et al. 2010, Anton 2020)[38,39] in two phases, from 1999 to 2004 and 2016 to 2020, and was tested using the same MAT protocols described above.

**Cougar overlap index**. Cougars and wolves generally select for different habitats due to their disparate hunting strategies[16]. However, due to YNP's high landscape heterogeneity and common use of the same prey populations, the two species exhibit variable overlap and interspecific encounters, including mortalities, occur[16]. Cougars and wolves range throughout YNP at low densities except for northern YNP where both occur at higher densities[39–41]. This spatial overlap with a definitive host species has been hypothesized to increase transmission to intermediate hosts[18]. Anton (2020) used non-invasive genetic surveys to sample northern YNP

and constructed a spatial model of cougar density within the sampled area[42], identifying cougar density per 3.2 km$^2$ cell (for specific methods see Anton 2020)[39]. We used this spatial model to determine areas within the northern range of YNP where cougar density was ≥1.8 cougars/100 km$^2$. This served as the cutoff between above and below-average cougar density[39,41]. The remainder of YNP, including areas outside of Anton 2020's sampling area, were classified as below average cougar density (less than 1.8/100 km$^2$) based on a paucity of cougar sightings and trail camera captures (usually <1 per year) and little to no use by GPS-collared cougars. In addition, the area was not conducive to year-round cougar occupancy because of the relatively flat, treed terrain with little relief and few canyons, and prey species only using the area from approximately June to September due to high elevation and severe winters[16,41]. It is possible there were a few small areas that were classified as below average density that have seasonal cougar densities approaching average levels. We calculated an index for each wolf's overlap with cougar density ≥1.8/100 km$^2$ by plotting each wolf territory, measured as a 95% minimum convex polygon (MCP) each biological year (April 1 to March 31) from aerial locations of wolf packs from 2000 to 2021, with the cougar spatial model. Although Anton's spatial model was built on individual genetic detections in winter (January-March) from 2014–2017, cougar use of northern YNP has remained relatively consistent throughout nearly three decades of wolf research conducted in YNP[16,39,42]. For each wolf tested, we averaged the annual overlap with cougar density ≥1.8 cougars/100 km$^2$ across years from birth to the time of the *T. gondii* test. Therefore, each wolf was assigned an average cougar overlap percent (mean = 31.53, range 0–100). As the individual wolf overlap percentages with cougar density were not evenly distributed, this index was divided into three categories: Low Cougar Overlap (LCO, 83 wolves with 0.0–5.0% overlap), Moderate Cougar Overlap (MCO, 83 wolves with 5.1–42.0% overlap), and High Cougar Overlap (HCO, 84 wolves with 42.1–100% overlap). This method ensured that the wolf overlap index with cougars was a metric of relative overlap compared to other wolves, not absolute wolf overlap with cougar density.

**Behavioural data collection**. Wolf packs were directly observed at varying frequency (approximately 20 to 250 times per year) depending on sightability within a pack's territory[43] from 1995 to 2021. Individual wolves were scored as a pack leader after meeting both of the following criteria: repeated observations (at least five times per year) of dominance over same-sex pack members and evidence of a pair bond (double scent-marking)[44] with the opposite-sex dominant individual. Wolves were recorded as habituated (1) if they approached or travelled near people or vehicles to within 10 meters. If a wolf was never recorded approaching within 10 m during the study period, it was coded 0. Dispersal was assessed post hoc when a wolf was tracked separately from its pack mates and away from the pack's territory and did not return. Cause of death was determined through necropsies in the field or at a laboratory.

## Statistics and reproducibility

**Demographic analysis**. We report seroprevalence for specific demographic categories (e.g., male versus female) and compare these proportions using z-scores. To examine individual variation in *T. gondii* infection, we used a generalized linear mixed model (GLMM) with a binomial distribution using *T. gondii* infection as the response variable. We compared a null model to a full model and report model performance based on DAIC and model weights ($w_i$). We used the program STATA for all analyses.

The full model included sex (either male or female, based on external reproductive organs), coat color (black or gray), and social status (leader or subordinate) at time of capture as categorical variables and age as a continuous variable. Studies on other species suggest that *T. gondii* seroprevalence and susceptibility may be influenced by certain hormones[19] and we included the variables sex, coat color, and social status because they are associated with certain hormonal patterns[45]. Gray wolves have higher cortisol levels compared to black wolves[23,24]. Social status influences, and is possibly influenced by, glucocorticoids[21,46]. Age was included to account for accumulating exposure risk throughout a wolf's life. We also included a cougar overlap index for each wolf. The cougar overlap index was averaged across years per individual, from birth until the time of the wolf's sample was taken. To account for re-tested individuals, and for unmeasured variables associated with individual wolves, we included wolf identity (WOLF_ID) as a random intercept. Year was not included as a random intercept because *T. gondii* exposure causes a chronic infection and it was unknown when seropositive wolves were exposed to *T. gondii*. Accounting for variation in year, relating to serostatus, would be only a reflection of capture effort in a given year rather than wolf exposure to *T. gondii*.

We compared a NULL model (testing if seroprevalence is best explained by the random intercept only) to a full model which included SEX, AGE, COAT COLOR, SOCIAL STATUS, and COUGAR OVERLAP. We tested for correlations between these variables and found none to be significant with p-values less than 0.05 (Supplementary Table 1). Variables from the best-performing model were significant if the p-value was 0.05 or less and the 95% confidence interval did not overlap zero. We constructed fitted value plots for the predicted probability that a wolf was seropositive for *T. gondii* antibodies based on the best-performing model.

**Behaviour analysis**. We identified three wolf behaviours considered risky: (1) dispersing, (2) becoming a pack leader, and (3) showing habituated behaviour around humans. We also identified two causes of death associated with known types of risk: (a) intraspecific mortality and (b) anthropogenic mortality which included deaths from hunting, poaching, vehicle strikes, lethal removal, and capture-related mortalities. We first report seroprevalence for each behavior or cause of death (e.g., percent of dispersers versus non-dispersers that were seropositive) and compare these proportions using z-tests. We then used generalized linear mixed models (GLMMs) with a binomial distribution where wolves displaying the behavior of interest were coded as a 1 and those that did not display the behavior were coded as a 0 (e.g., 1=dispersed or 0=did not disperse). We developed a base model for each behaviour and included covariates associated with the specific wolf behaviour and decision-making[15]. We then compared this to a model that included all the same variables plus *T. gondii* seroprevalence. We report model performance based on DAIC and model weights ($w_i$). We included wolf identity (WOLF_ID) as a random intercept and used the program STATA for all analyses.

To test whether *T. gondii* infection affected dispersal, we compared two models. $DISP_1$ included SEX, SYSTEM, TIME AVAILABLE, and TOXO status. $DISP_2$ included all the same variables minus TOXO. This method allowed us to determine the effect of *T. gondii* while controlling for other important factors, described below. The same method was used to test for *T. gondii* effects on leadership: $LEAD_1$ included SYSTEM, TIME AVAILABLE, and TOXO and was compared to $LEAD_2$ including only SYSTEM and TIME AVAILABLE. The models testing for effects on habituation included: $HAB_1$—SEX, TIME AVAILABLE, SYSTEM, and TOXO and $HAB_2$—SEX, TIME AVAILABLE, and SYSTEM. For both GLMMs related to cause of death we compared $INTRA$-$MORT_1$ and $ANTHRO$-$MORT_1$ including SEX, AGE CLASS, SYSTEM, and TOXO to $INTRA$-$MORT_2$ and $ANTHRO$-$MORT_2$ including SEX, AGE CLASS, and SYSTEM. We included individual wolf ID as a random intercept for all models except the habituation analysis ($HAB_1$ and $HAB_2$) which were restricted by sample size and models attempting to account for repeated measures did not converge.

To account for the duration a wolf was monitored while infection status was known, we used the continuous variable TIME AVAILABLE. Temporal measure TIME AVAILABLE was the total number of days each wolf was monitored (mean: 746, range: 3–3800). The focal time period for the response (i.e., risky behaviour) was dependent on the time period of a wolf's known *T. gondii* serostatus. Wolves that were seronegative had a TIME AVAILABLE from the time of their test backwards to their birth, and wolves that were seropositive had a TIME AVAILABLE from their test date forward to death, or to present time if still alive. Restricting observed behaviours to the time period for which we knew a wolf's infection status ensured that we accurately accounted for parasite infection when assessing behaviour. For wolves that tested negative, we subtracted 180 days from their time available because the first six months of a wolf's life is spent as dependents at a den or rendezvous site, and they are not old enough to disperse or become leaders.

SEX (male or female) and AGE CLASS (pup, yearling, adult, or old) were included in a subset of the behavior models to control for their known effects on wolf life history, particularly related to dispersal, where males are more likely to disperse and dispersal rates increase with age, and survival[25].

Relatively high wolf density in northern YNP may affect wolf behavior. For example, wolves that lived in northern YNP had closer selection distance to roads[47] and this may correlate with habituated behavior. High wolf density may also affect behavior such as dispersal given the greater access to potential mates. To account for this dynamic, we included SYSTEM (northern YNP or non-northern YNP) in the behavior models.

Variables from the best-performing model were significant if the p-value was 0.05 or less and the coefficient 95% confidence interval did not overlap zero. If the best-performing model for each behavior included TOXO, we constructed fitted value plots to visualize the predicted probability of the behavior given serostatus and the other significant variables.

**Reporting summary**. Further information on research design is available in the Nature Research Reporting Summary linked to this article.

## Data availability
The datasets generated and analyzed during the current study are available from the corresponding author upon request. Source data are available in Supplementary Data.

## Code availability
The code used during the analysis of the current study are available from the corresponding author upon request.

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

## Acknowledgements

We thank Rick McIntyre and the many volunteers and visitors to Yellowstone National Park who helped make this project possible. We are deeply appreciative to Dr. Mark Hebblewhite and Dr. Doug Emlen for their valuable insight and feedback on this manuscript. We additionally want to thank Brenna Cassidy and Wes Binder for their perspective and input on the topic. This work could not be done without the safe piloting of Mark Packila from Wildlife Air, Jim Pope from Leading Edge, Bob Hawkins from Hawkins and Powers, Inc., Roger Stradley and Steve Ard from Gallatin Flying Service, and Stephan Robinson and Grayson Sperry of Ridgeline Aviation. We would like to thank Toni Ruth, Hornocker Wildlife Institute, and Wildlife Conservation Society for providing cougar serum samples from northern Yellowstone during 1999–2004. We also thank significant donors to the Yellowstone Wolf Project: Valerie Gates, Annie and Bob Graham, and Frank and Kay Yeager. Thanks to the Cornell University Animal Health Diagnostic Center, as well as the U. S. National Park Service, U. S. Geological Survey, Yellowstone Forever, and the National Science Foundation (DEB- 0613730 and DEB-1245373) for their support of this work. Any mention of trade, firm, or product names is for descriptive purposes only and does not imply endorsement by the U.S. government.

## Author contributions

C.J.M., K.A.C., and E.E.S. conceived of the research questions. E.E.B. and C.B.A. provided conceptual advice and D.W.S. and D.R.S. provided conceptual advice and supervised the project. D.W.S., D.R.S., E.E.S., K.A.C., C.J.M, and C.B.A. collected physical samples while K.A.C., E.E.S., C.B.A., D.R.S., D.W.S., and E.E.B. collected observational data. K.A.C. analyzed data and created the schematics and figures. C.J.M. and K.A.C. wrote the paper with input from the other authors.

## Competing interests

The authors declare no competing interests.
