## [Peer Review File · Communications Biology]

Reviewers' comments:

Reviewer #1 (Remarks to the Author):

The manuscripts reports a very interesting study aiming at assessing 1. The factors affecting the presence of *T.gondii* infection in wild wolves and 2. The potential effect of *T.gondii* infection on wolves' risk-taking behavior.

Given the paucity of studies on the subject, the highly intriguing relationship between infection, behavior and potential ecological implications at a broader level (which the authors speculate about in the discussion), I find the study original and highly relevant, and I recommend it for publication

There however a number of aspects which I think need addressing before publication, mostly to help the reader understand the main points and make the study more incisive.

I also have some statistical concerns, which I will outline here. I realize there are differences in such approaches across fields, so I voice my concerns below, however I am also open to the editor taking a different view.

Introduction

The study has two aims

1. understand what affects infection rate and
2. evaluate if infection affects behaviour

Whereas the second aim is nicely introduced, referring to the few studies available the first aim is not. I come from a behaviour background, and don't know much about *T.gondii*, so for me, whereas it is very clear why the predictor 'cougar overlap' would affect presence of infection in wolves, it is not so clear why the other variables (sex, age, status and coat colour) would predict its presence.

My suggestion would be to re-work the intro a bit so as to introduce both aims, state the aims more explicitly at the end of the intro and more clearly link the background literature to the predictors. This will help the reader and make the paper more incisive.

Similarly, with the effect of TOXO on behavior, there are a set of control predictors which need to be included because they are known to affect risk-taking behavior, for example sex (male are less neophobic than females, and show higher dispersal than females) and age (younger are more risk-prone than older). A sentence to introduce these will then help the reader with why you chose specific predictors to be included in your analyses.

Minor

LI 56: the reference to Gering's et al, is unclear since the reference method does not allow to see directly which study you are referring to. consider re-phrasing.

LI 77: does this mean the social status was inferred by hormonal levels? phrasing is unclear here.

LI 83: although it becomes clear later in the paper that both these causes of death are considered 'high risk', from this sentence alone it is not so clear. Please rephrase.

Study area: The north south division of YNP, is then used as a predictor variable for some of the models (SYSTEM). Here it would be important to describe a bit more the features of the two areas which are thought to differ, which are relevant for the analyses, e.g. human presence/roads etc?

LI 127: I am not clear if the 'equivocal' were treated as a third category in the analyses. Can you rephrase please.

Behavioural data:

Quite some information is missing in this section.

LI 164: observed how? directly? camera traps?

LI 166: determination of pack leaders: 'repeated observations'? how many? and what kind of behaviours were coded? Did both criteria have to be fulfilled (so both within same sex behaviour and double-scent marking) or just one was enough to be established as a pack leader? and presumably referring to the same year of capture (and hence toxo analyses)?

LL168: categorization of 'habituated' to human/cars is unclear. how often did they have to approach or travel near people to be considered habituated? once? how was this normalized by the number of times they 'could' display such behaviours? (so occasions in which a person or vehicle was available for them to approach).

Demographic analyses

LI177: so yes/no infected vs not infected correct? so the equivocal results' were dropped from analyses? Please state this more clearly and include how many were dropped.

L:178-185: an explanation of the link between hormones and prevalence of *T.gondii* should be in the intro, so that the reader understands why you are including this as a control variable. I sentence is sufficient, but important. You also need to clarify in the intro that you use, sex and coat colour as proxies for hormonal levels, and explain the 'hormonal patterns' you refer to here including references. The expectation would be that you measured hormones in your samples as well (no chance you have hair or other material in which you could get baseline measures of glucocorticoids and testosterone?), so you need to justify the use of the proxies instead.

In relation to this a few issues come to mind. Status is likely linked to both higher glucocorticoid and testosterone levels, and they may or may not co-vary with coat colour. So you need to carry out a preliminary analyses to check the extent that these predictor variables co-vary. If they do not, I am fine keeping all of them in the model. If not please explain which predictors co-vary and then make a justified choice of which to include in the model (the one you think is most relevant).

Please also specify how each predictor was entered in the model: age was categorical (if so how many levels) or continuous? Coat colour (2-level: black grey) etc.

Models:

I don't understand the rationale for this modelling approach. By carrying out multiple models on the same data you are increasing your chances of finding false positives.

Please see two references in this regard:

Mundry 2011 Issues in information theory-based statistical inference—a commentary from a frequentist's perspective. *Behav Ecol Sociobiol* (2011) 65:57–68
DOI 10.1007/s00265-010-1040-y

Roger Mundry and Charles L. Nunn (2009) Stepwise Model Fitting and Statistical Inference: Turning Noise into Signal Pollution. *The American Naturalist*, Vol. 173, No. 1 pp. 119-123

Please carry out a full-null model comparison, with all your predictor variables in one model (having checked that they do not co-vary), and including your interaction factor (explaining why you need it). If the interaction factor is not sig. please drop it and re-run model with all your predictors. Then results of this model should be reported in full. No other models are needed.

Behaviour analyses:

Similarly to my comment above in the intro, it is important to introduce why certain control predictors

are included (sex: males are generally less neophobic and disperse more, age: younger animals are most risk-prone than older ones, all this is available in the animal literature, if not always wolves) etc.

I would also suggest to think a bit more about whether for some models other control factors may be necessary. For example, in areas with higher human presence, the habituation to cars/people may be different. Was this controlled for? how? is this your SYSTEM variable? please clarify this. If indeed North vs. South YNP differ in this aspect it should be described better in the Study area, and clarified here. An alternative possibility is to create a 'human activity' variable for each pack based on road coverage, or use by tourists.

Similarly, the likelihood of encountering a road/human is also going to affect the likelihood of human induced death and intraspecific death is going to be affected by pack encounter rates, and from a previous study by the group, from the size difference of the attacking and attacked pack. So I think it would be necessary that for each model 1) dispersal 2) pack leadership 3) habituation and 4) cause of death, the authors think about the control variables that would be needed. These may not all be the same across models: for example it is not clear to me why SYSTEM would affect whether one become a leader or not, or chooses to disperse. Do the north south areas of YNP differ in the social structure of the wolves? Is it more likely that a wolf will become a leader or disperse in one of the two areas? If so then please explain this so that the rationale for including this variable in the model is clear to the reader.

Including control predictors thought to influence the behaviour where possible will greatly strengthen the results of the paper, as will explaining the rationale for their inclusion.

As for the stats analysis pertaining to aim1, I would argue that also for aim2 running so many models is not just redundant but inappropriate (see references provided).

Indeed one may even argue that if the question is whether toxo increases risk-taking behaviour, above and beyond the other variables already known to affect risk-taking behaviour (e.g. sex, age etc), then the null model should include all these control variables, and the full model should have all the control variable plus the predictor of interest: TOXO.

I am also fine however, with running the full model compared to the null-intercept only model. BUT all control and predictor variable need to be included in one model, and only one model should be run.

Results

Aim 1: factors affecting presence of TOXO

The descriptive results are very useful. However, at one point it gets a bit confusing in relation to what dataset is then used for the analyses. I would suggest some re-structuring. Cougars first, to get them out of the way, then descriptive of the TOXO presence in wolves. Then a clear statement of n. of samples over which years used for the model. And results of the model.

LI 257: how you categorized age should go in the 'analyses section' not here in the results.

LI 287: I realize the approaches to modelling vary amongst field of research. In my field (behavior), running this many models on the same dataset and applying a model selection approach based on AIC is considered a practice which increases the chances of false positives (see references). I share these concerns. What emerges here is that the important predictor variable is 'cougar overlap', if this is taken into account, other predictors such as age are unimportant. If each predictor is taken separately other effects emerge, but I would not consider them solid results, since data is still coming from area with cougar overlap which will therefore always have to be included in the analyses.

LI314: 'non-significant' not 'insignificant'

Aim 2: I think most of the comments I made to to the analyses, make any comments to the results redundant.

The graphs and figures are all very informative, as are the captions.

Discussion

I found the discussion very interesting since it comes from multiple perspectives and includes the broader implications from an ecological perspective. I have just one small comment.

LI 474: novel situation I understand.. but why would they seek out felid scent? This needs a bit more explanation.

thanks for a really interesting piece of work. despite my many comments it was a pleasure to read, I hope this suggestions above can improve its incisiveness and clarity.

best regards
Sarah Marshall-Pescini

Reviewer #2 (Remarks to the Author):

This is a potentially very interesting piece of work, displaying unique data collected for several years in a unique and invaluable ecological context. I certainly appreciated the overall story, the infographics, and the theoretical background.

Before being able to judge this paper in a fair way, however, I would need to better understand the analytical part of it, because without robust analyses it is very difficult to verify whether authors are able to draw final conclusions from their results.

Data analyses have been carried out with STATA, so, differently to colleagues working in R and sharing data and codes, the Authors here must spend extra effort to convince the readers about their analysis. In short, full transparency is needed. The method section, however, is the weakest part of this paper. Analyses are explained poorly, and I suppose that most readers would struggle to understand and reproduce the same analysis:

- there is no mention of GLMM diagnostics, whether model assumptions are met
- there is no mention of spatial and temporal autocorrelations, whether these have been accounted for, or whether significance patterns are affected by these issues
- in the method section, there is no mention of the multi-model comparison strategy (e.g., AIC or other similar methods) – they are used in the results, but not mentioned in the methods
- in the result sections, there are statistical tests not mentioned in the methods (lines 257-262)
- the method section explaining alternative model structures is so confusing: why did you use a multi-model comparison in the first place? Why didn't you use a priori model structures considering your well-defined hypotheses? You need to justify your modelling approach to be credible to the readers
- model structures seem a bit weak too, e.g. why year of study is not included as random intercept beyond the ID of the individuals?

- the method section mentions 4 metrics but lines 203-208 only refers to the modelling of dispersal behaviour – readers would guess whether you have forgotten to explain the other 3 models.

To be fully appreciated, this paper needs a full rewriting of the method section, with a better clarification of sample sizes, modelling approaches, model assumptions, and more transparency on the results, including output from STATA as supplementary material. The rest of the paper is simply great, this is a potentially outstanding story, but science needs robust empirical evidence first.

Dear Communications Biology Editor, Reviewer 1, and Reviewer 2,

Thank you for your time and effort to review this manuscript titled "Parasitic infection increases risk-taking in a social, intermediate host carnivore." We appreciate your careful consideration to examine this work in detail and to respond with excellent suggestions to improve the manuscript. We have worked to address the concerns from the reviews, which includes a new analysis approach for both of the paper's aims, a complete rewriting of the Methods sections, substantial reworking of the Introduction and Results, as well as general edits for clarity and flow throughout the manuscript. We address each issue raised by Reviewers 1 and 2 in detail below (blue bold).

Please note that we added a coauthor to this paper (Colby B. Anton) as his contribution was critical to the new version of this manuscript.

Reviewers' comments:

Reviewer #1 (Remarks to the Author):

*The manuscript reports a very interesting study aiming at assessing 1. The factors affecting the presence of *T.gondii* infection in wild wolves and 2. The potential effect of *T.gondii* infection on wolves' risk-taking behavior.*

Given the paucity of studies on the subject, the highly intriguing relationship between infection, behavior and potential ecological implications at a broader level (which the authors speculate about in the discussion), I find the study original and highly relevant, and I recommend it for publication

There however a number of aspects which I think need addressing before publication, mostly to help the reader understand the main points and make the study more incisive.

I also have some statistical concerns, which I will outline here. I realize there are differences in such approaches across fields, so I voice my concerns below, however I am also open to the editor taking a different view.

Thank you for your support of the original nature of this work. We certainly want to make this work more incisive and understandable to readers. We appreciate the detail put into describing your concerns, especially with the statistical analysis, and believe we have addressed your comments in this new version of the manuscript. We believe these changes have improved the work substantially. Our specific responses are below.

Introduction

The study has two aims

- 1. understand what affects infection rate and*
- 2. evaluate if infection affects behaviour*

*Whereas the second aim is nicely introduced, referring to the few studies available the first aim is not. I come from a behaviour background, and don't know much about *T.gondii*, so for me, whereas it is very clear why the predictor 'cougar overlap' would affect presence of infection in wolves, it is not so clear why the other variables (sex, age, status and coat colour) would predict its presence.*

My suggestion would be to re-work the intro a bit so as to introduce both aims, state the aims more explicitly at the end of the intro and more clearly link the background literature to the predictors. This will help the reader and make the paper more incisive.

We rewrote large portions of the Introduction to address this suggestion. We added a clear statement of our two aims (Lines 75-103) and added a paragraph of previous studies to justify the inclusion of the demographic factors in the analysis determining which factors influence *T. gondii* seroprevalence.

Text added/edited: *“Our first aim was to determine which demographic and ecological factors affect *T. gondii* infection in wolves in Yellowstone National Park. We tested individual demographic factors including age, sex, social status at time of capture, and coat color due to their potential variation in disease susceptibility. Previous research has found the risk of *T. gondii* infection increases with age due to accumulating risk of exposure with time.^{17,18} The other three wolf demographic factors were included because of their links to certain hormones, which may influence an animal's susceptibility to infection.¹⁹ Sex hormones play a role in infection risk and, once infected, hormone production is altered for some species¹⁹; however, other studies found no link between *T. gondii* seroprevalence and sex.^{14,17,18} Due to natural variations in hormone levels (testosterone, progesterone, estrogen, etc.) between the sexes,²⁰ there may be differing risks and subsequent behavioral responses to infection.*

*Previous research has found social status (e.g., pack leaders)²¹ and coat color (gray coat color wolves have higher cortisol levels and increased behavioral aggression)²² linked to varying hormone levels and immune defense.^{23,24} To determine if seroprevalence is affected by the amount of spatial overlap with a *T. gondii* definitive host (i.e., cougars), we included an overlap index for each wolf and areas of high cougar density.*

*Our second aim was to determine if *T. gondii* infection influences wolf behavior. We identified three wolf behaviours associated with greater risk-taking: (1) dispersing from a pack, (2) achieving dominant social status (referred to as becoming a leader), (3) approaching people or vehicles (referred to as habituation), and two causes of death associated with increased risk:(a) intraspecific mortality, death by other wolves through interpack fights, or (b) anthropogenic mortality, death by humans due to decreased proximity to humans or human structures. As behavior can be influenced by many factors, we controlled for certain variables in each of the behavior models: sex can influence behaviors such as dispersal, and age can influence the probability of a certain behavior occurring.²⁵ Northern YNP has very high wolf density, the roads are open year-round, the elevation is lower and provides winter range for ungulates and opportunities for wolf hunters just outside the park boundary. All these factors may affect wolf behavior as the wolves there may have increased opportunities to disperse, to die, and may be more susceptible to habituation. Therefore, we controlled for YNP system (northern or not) as well. In controlling for these factors that may influence wolf behavior, we aim to isolate the influence of *T. gondii* infection on behavior. We tested if serostatus influenced the odds of a wolf performing these behaviors or dying of one of these causes.”*

Similarly, with the effect of TOXO on behavior, there are a set of control predictors which need to be included because they are known to affect risk-taking behavior, for example sex (male are less neophobic than females, and show higher dispersal than females) and age (younger are more risk-prone than older). A sentence to introduce these will then help the reader with why you chose specific predictors to be included in your analyses.

We added a paragraph, Lines 195-199, describing the reasons to control for certain factors in the behavior analysis and expanded on our decision-making process related to these variables in the Introduction and Methods/Data Analysis sections.

Edited text: “Studies on other species suggest that *T. gondii* seroprevalence and susceptibility may be influenced by certain hormones¹⁹ and we included the variables sex, coat color, and social status because they are associated with certain hormonal patterns.³⁷ Gray wolves have higher cortisol levels

compared to black wolves.^{23,24} Social status influences, and is possibly influenced by, glucocorticoids.”
The lines pertaining to this in the Introduction were 75-103 (pasted above).

Minor

Ll 56: the reference to Gering's et al, is unclear since the reference method does not allow to see directly which study you are referring to. consider re-phrasing.

*We rephrased this and smoothed out those few sentences for clarity, Lines 59-64 (“One of the few studies focused on infection impacts on behavior in a wild mammal, Gering et al. (2021) found that toxoplasmosis was associated with increased boldness in hyena (*Crocuta crocuta*) cubs and that seropositive hyenas of all ages were more likely to be killed by African lions (*Panthera leo*).¹⁴ That study demonstrated a mechanistic link between toxoplasmosis and an individual's fitness through behaviour and decision-making.”)*

Ll 77: does this mean the social status was inferred by hormonal levels? phrasing is unclear here.

*We weren't able to collect hormone data from the wolves in this study. Because social status and hormone levels could covary (although for wolves this is unclear), we included an explanation in Introduction (Lines 79-86) to describe the reasons for this decision: *T. gondii* not only affects hormones but some hormones can make certain animals more or less susceptible to infection. We included sex, coat color, and social status in this analysis because of their variation in hormone levels (male vs female) to see if there were differences in *T. gondii* infection rates that might be explained by hormones differences. None of these variables ended up significant in the best-model. New/edited text: “The other three wolf demographic factors were included because of their links to certain hormones, which may influence an animal's susceptibility to infection.¹⁹ Sex hormones play a role in infection risk and, once infected, hormone production is altered for some species¹⁹; however, other studies found no link between *T. gondii* seroprevalence and sex.^{14,17,18} Due to natural variations in hormone levels (testosterone, progesterone, estrogen, etc.) between the sexes,²⁰ there may be differing risks and subsequent behavioral responses to infection. Previous research has found social status (e.g., pack leaders)²¹ and coat color (gray coat color wolves have higher cortisol levels and increased behavioral aggression)²² linked to varying hormone levels and immune defense.^{23,24}”*

Ll 83: although it becomes clear later in the paper that both these causes of death are considered ‘high risk’, from this sentence alone it is not so clear. Please rephrase.

Thank you for this suggestion, we have rephrased the previous line regarding propensity of risk and cause of death. (Lines 92 to 94)

Edited/added text: *“and two causes of death associated with increased risk:(a) intraspecific mortality, death by other wolves through interpack fights, or (b) anthropogenic mortality, death by humans due to decreased proximity to humans or human structures.”*

Study area: The north south division of YNP, is then used as a predictor variable for some of the models (SYSTEM). Here it would be important to describe a bit more the features of the two areas which are thought to differ, which are relevant for the analyses, e.g. human presence/roads etc

As behavior can be influenced by many factors, we controlled for certain variables in each of the behavior models: sex can influence behaviors such as dispersal, and age can influence the probability of a certain behavior occurring.²⁵ Northern YNP has very high wolf density, the roads are open year-round, the elevation is lower and provides winter range for ungulates and opportunities for wolf hunters just outside the park boundary. All these factors may affect wolf behavior as the wolves there may have increased opportunities to disperse, to die, and may be more susceptible to habituation. Therefore, we controlled for YNP system (northern or not) as well. In controlling for these factors that may influence wolf behavior, we aim to isolate the influence of *T. gondii* infection on behavior.”

New text in Methods (Lines 160-162): *“In addition, the area was not conducive to cougar year-round occupancy because of the relatively flat, treed terrain with little relief and few canyons, and prey species only using the area from approximately May to October due to high elevation and severe winters^{16,33”}*

In addition, Lines 251-257: *“Relatively high wolf density in northern YNP may affect wolf behavior. For example, wolves that lived in northern YNP had closer selection distance to roads³⁹ and this may correlate with habituated behavior. High wolf density may also affect behavior such as dispersal given the greater access to potential mates. To account for this dynamic, we included SYSTEM (northern YNP or non-northern YNP) in the behavior models.”*

Ll 127: I am not clear if the 'equivocal' were treated as a third category in the analyses. Can you rephrase please.

We treated equivocal test results as negative tests in the analysis section. We moved this from Line 126 to Line 140 for better flow and clarity.

Behavioural data:

Quite some information is missing in this section.

Ll 164: observed how? directly? camera traps?

We added the word “directly” to clarify how wolf observations occurred (Line 179).

Ll 166: determination of pack leaders: 'repeated observations'? how many? and what kind of behaviours were coded? Did both criteria have to be fulfilled (so both within same sex behaviour and double-scent marking) or just one was enough to be established as a pack leader? and presumably referring to the same year of capture (and hence toxo analyses)?

We added more details to this description in the Methods section (Lines 179-187). Repeat observations were ≥ 5 per year, behaviors were raised tail/ears during socialization and dominating same-sex pack mates. Both criteria had to be filled (same-sex interactions and double-scent marking with opposite-sex mate). For some context apart from the added/edited text: Pack leaders often stayed leaders for many years until their death. After a death we would put extra effort into figuring out which wolf was the new leader. For brevity, and as the change in leadership was rare, we only added the number of observations and both criteria to the main text.

LL168: categorization of 'habituated' to human/cars is unclear. how often did they have to approach or travel near people to be considered habituated? once? how was this normalized by the number of times they 'could' display such behaviours? (so occasions in which a person or vehicle was available for them to approach).

This data is admittedly coarse and not possible to normalize in a wild system, although we did try to account for some of those issues in the other factors in the models (system, time). Lines 183-185 describe this variable specific to the distance needed for a wolf to be classified as “habituated” and the categorization method (binary 1/0). For background, we keep records of wolves that display habituated behavior and one of the items on the dataform is “How close to humans or vehicles did wolves approach or travel?” For this study we filtered out any wolves getting closer than 10 meters and gave those wolves a 1. Any wolves that did not have a record of this behavior were given a 0. A wolf only had to have one of these dataform records in order to be given a score of 1. It is possible some of the wolves with scores of 0 had approached people/vehicles at some time, but habituation (to within 10 m) is fairly rare in YNP and any wolves getting that close are generally repeat offenders and well known by staff. We do not believe we recorded any false-positive habituated wolves and it is possible, yet unlikely, wolves were recorded as false-negative for habituation. A sliding scale of habituation would also be an appropriate way of examining this behavior, but this was not possible with this study given only a subset of wolves were tested for *T. gondii* and only a

few dozen have been habituated in the history of our record keeping, so the overlap was minimal. We believe we constructed an adequate variable given the data that are available to us.

Demographic analyses

L1177: so yes/no infected vs not infected correct? so the equivocal results' were dropped from analyses?

Please state this more clearly and include how many were dropped.

We now include a line clarifying that equivocal samples were treated as negative in the analysis (Line 140)).

L:178-185: an explanation of the link between hormones and prevalence of T.gondii should be in the intro, so that the reader understands why you are including this as a control variable. I sentence is sufficient, but important. You also need to clarify in the intro that you use, sex and coat colour as proxies for hormonal levels, and explain the 'hormonal patterns' you refer to here including references. The expectation would be that you measured hormones in your samples as well (no chance you have hair or other material in which you could get baseline measures of glucocorticoids and testosterone?), so you need to justify the use do the proxies instead.

We added these explanations and justifications for the links between hormones and the variables included in the model for aim 1 (sex, coat color, social status). We also added this information to the Introduction. Lines 75-103, 195-199.

In relation to this a few issue come to mind. Status is likely linked to both higher glucocorticoid and testosterone levels, and they may or may not co-vary with coat colour. So you need to carry out a preliminary analyses to check the extent that these predictor variables co-vary. If they do not, I am fine keeping all of them in the model. If not please explain which predictors co-vary and then make a justified choice of which to include in the model (the one you think is most relevant).

We added a table of correlations between the variables used in aim 1 and aim 2, submitted as Supplementary Information tables with this revised manuscript. None of the variables are significantly correlated. We added language in the Introduction explaining the reason for including social status (two different publications have found opposing relationships between social status and cortisol--Sands et al. 2004, Molnar et al. 2015) so that pattern is somewhat unknown in wild wolves), coat color (coat color in dogs and wolves is correlated with cortisol, Cassidy et al. 2017) as well as why infection may covary with a certain sex hormones.

Cassidy, K. A., Mech, L. D., MacNulty, D. R., Stahler, D. R. & Smith, D. W. Sexually dimorphic aggression indicates male gray wolves specialize in pack defense against conspecific groups. *Behavioural Processes* 136, 64–72 (2017).

Molnar, B. *et al.* Environmental and intrinsic correlates of stress in free-ranging wolves. *PLoS ONE* 10, 1–25 (2015).

Sands, J. & Creel, S. Social dominance, aggression and faecal glucocorticoid levels in a wild population of wolves, *Canis lupus*. *Animal Behaviour* 67, 387–396 (2004).

Please also specify how each predictor was entered in the model: age was categorical (if so how many levels) or continuous? Coat colour (2-level: black grey) etc.

We added details in Data Analysis about the type of measure used for each variable: categorical/continuous. Lines 195-197: “The full model included sex (either male or female, based on external reproductive organs), coat color (black or gray), and social status (leader or subordinate) at time of capture as categorical variables and age as a continuous variable.”

Models:

I don't understand the rationale for this modelling approach. By carrying out multiple models on the same data you are increasing your chances of finding false positives.

Please see two references in this regard:

*Mundry 2011 Issues in information theory-based statistical inference—a commentary from a frequentist's perspective. Behav Ecol Sociobiol (2011) 65:57–68
DOI 10.1007/s00265-010-1040-y*

Roger Mundry and Charles L. Nunn (2009) Stepwise Model Fitting and Statistical Inference: Turning Noise into Signal Pollution. The American Naturalist, Vol. 173, No. 1 pp. 119-123

Please carry out a full-null model comparison, with all your predictor variables in one model (having checked that they do not co-vary), and including your interaction factor (explaining why you need it). If the interaction factor is not sig. please drop it and re-run model with all your predictors. Then results of this model should be reported in full. No other models are needed.

Thank you for this suggestion. This is a different approach to model selection and comparison than what we originally intended for this project, but given the type of data, the lack of other research on this exact question, and your suggestion, we decided to change our original analysis approach

and apply the suggested approach instead. To do so, we rewrote the Methods/Data Analysis sections, specifically Lines 190-194 and 209-215, and reanalyzed the data using this approach—a comparison between a null and full model. Our results are described in Results section and displayed in a new table (Table 1). The new full model is slightly different from the former “best” model from our original draft because it includes coat color, although coat color is not a statistically significant variable in the model output.

Behaviour analyses:

Similarly to my comment above in the intro, it is important to introduce why certain control predictors are included (sex: males are generally less neophobic and disperse more, age: younger animals are most risk-prone than older ones, all this is available in the animal literature, if not always wolves) etc.

We made these additions and justifications in Introduction. Lines 75-103 and Lines 195-202.

I would also suggest to think a bit more about whether for some models other control factors may be necessary. For example, in areas with higher human presence, the habituation to cars/people may be different. Was this controlled for? how? is this your SYSTEM variable? please clarify this. If indeed North vs. South YNP differ in this aspect it should be described better in the Study area, and clarified here. An alternative possibility is to create a ‘human activity’ variable for each pack based on road coverage, or use by tourists.

We added language to the Introduction to better describe the difference between northern YNP and non-northern YNP. There is additional language in Data Analysis section. Because the addition of SYSTEM is based on study area factors but seemed to us to be related to an analysis decision, we felt it fit better in Intro/Data Analysis explanations than Study Area descriptions. We added and edited text at Lines 94-102, 160-162, and 253-257.

Similarly, the likelihood of encountering a road/human is also going to affect the likelihood of human induced death and intraspecific death is going to be affected by pack encounter rates, and from a previous study by the group, from the size difference of the attacking and attacked pack. So I think it would be necessary that for each model 1) dispersal 2) pack leadership 3) habituation and 4) cause of death, the authors think about the control variables that would be needed. These may not all be the same across models: for example it is not clear to me why SYSTEM would affect whether one become a leader or not, or chooses to disperse. Do the north south areas of YNP differ in the social structure of the wolves? Is it more likely that a wolf will become a leader or disperse in one of the two areas? If so then please explain this so that the rationale for including this variable in the model is clear to the reader.

Including control predictors thought to influence the behaviour where possible will greatly strengthen the results of the paper, as will explaining the rationale for their inclusion.

The variable SYSTEM is intended to capture and control for some of the variation in risk available to wolves. We added language in the Introduction (Lines 94-102) about northern YNP having more wolves, roads, and humans--all factors that could not only impact behavior but also increase the likelihood of death. SYSTEM is included in each of the behavior and death models for this reason. It is included in the dispersal and leadership models because of the very high wolf density in northern YNP and the increased opportunity for wolves living in the area to encounter non-pack mates and attempt to start a new pack (both dispersal and potentially leadership opportunity). In addition, pack turnover in northern YNP is much higher than non-northern YNP, creating openings for new packs to form from dispersers, along with new leaders. Wolves living in non-northern YNP do not have these same opportunities, even though there is danger that comes along with meeting non-pack mates, it seems that it may trigger behaviors such as dispersal and becoming a leader. We added wording to help clarify this in the Introduction (94-102) and Data Analysis (Lines 160-162, 253-257).

As for the stats analysis pertaining to aim1, I would argue that also for aim2 running so many models is not just redundant but inappropriate (see references provided).

Indeed one may even argue that if the question is whether toxo increases risk-taking behaviour, above and beyond the other variables already known to affect risk-taking behaviour (e.g. sex, age etc), then the null model should include all these control variables, and the full model should have all the control variable plus the predictor of interest: TOXO.

I am also fine however, with running the full model compared to the null-intercept only model. BUT all control and predictor variable need to be included in one model, and only one model should be run.

Thank you for this suggestion. Your model selection suggestion works much better and is a smoother, simpler way to stay focused on *T. gondii* while still controlling for the appropriate variables that may impact behavior/death. We completed a new analysis using this method: a comparison of two models for each behavior, one with all of the control variables and one with all of the control variables plus *T. gondii*. We dropped all of the univariate models and added a table of the variables and correlations to the SI (SI Table 1). Reassuringly, all of the results for the behavior analyses were the same as the previous analysis. We did edit the Results section throughout to account for the new model output (Tables 2 and 3) but the best/better performing models were the

same and predictions (Figures 3 and 4) based on that models did not change.

Results

Aim 1: factors affecting presence of TOXO

The descriptive results are very useful. However, at one point it gets a bit confusing in relation to what dataset is then used for the analyses. I would suggest some re-structuring. Cougars first, to get them out of the way, then descriptive of the TOXO presence in wolves. Then a clear statement of n. of samples over which years used for the model. And results of the model.

Thank you for the suggestion. We made the suggested edits and brought the information about *T. gondii* seroprevalence in cougars to the beginning as evidence of cougars being the definitive host in this system (Lines 267-270). We then clarified the order of Results. We kept Aim 1 and Aim 2 separate so the order is: cougar information, descriptive of Aim 1, GLMMs of Aim 1, descriptive of Aim 2, GLMMs of Aim2.

Ll 257: how you categorized age should go in the 'analyses section' not here in the results.

Thank you, we added this to analyses section (Lines 196-197).

Ll 287: I realize the approaches to modelling vary amongst field of research. In my field (behavior), running this many models on the same dataset and applying a model selection approach based on AIC is considered a practice which increases the chances of false positives (see references). I share these concerns. What emerges here is that the important predictor variable is 'cougar overlap', if this is taken into account, other predictors such as age are unimportant. If each predictor is taken separately other effects emerge, but I would not consider them solid results, since data is still coming from area with cougar overlap which will therefore always have to be included in the analysis

Great points and we address these issues by changing the analysis framework/approach to that suggested above.

Ll314: 'non-significant' not 'insignificant'

We changed this word as suggested.

Aim 2: I think most of the comments I made to to the analyses, make any comments to the results

redundant.

The graphs and figures are all very informative, as are the captions.

Discussion

I found the discussion very interesting since it comes from multiple perspectives and includes the broader implications from an ecological perspective. I have just one small comment.

Ll 474: novel situation I understand.. but why would they seek out felid scent? This needs a bit more explanation.

We included a few word changes in the Discussion to clarify this (Lines 475-476). An intermediate host seeking out felid scent gives the parasite a better chance of completing its life-cycle because it needs to infect a definitive host; more specifically, felid prey that approach felids via scent attraction may be more likely to be predated and consumed by the felid. While this phenomenon likely evolved in common felid prey (e.g., rodents), the same behavior was found in chimpanzees, so the mechanism may still exist in non-prey items even though it would rarely lead to predation of the intermediate host. We included references for infected chimpanzees and rats have been shown to seek out and spend time near felid urine, as uninfected individuals who avoid the feline urine and spend more time near “control” urine samples, which are non-predator species urine samples. The Gering et al. (2021) work with hyenas also suggested this dynamic where intermediate hosts sought out or approached a felid.

thanks for a really interesting piece of work. despite my many comments it was a pleasure to read, I hope this suggestions above can improve its incisiveness and clarity.

best regards

Sarah Marshall-Pescini

Reviewer #2 (Remarks to the Author):

This is a potentially very interesting piece of work, displaying unique data collected for several years in a unique and invaluable ecological context. I certainly appreciated the overall story, the infographics, and the theoretical background.

Before being able to judge this paper in a fair way, however, I would need to better understand the analytical part of it, because without robust analyses it is very difficult to verify whether authors are able to draw final conclusions from their results.

Data analyses have been carried out with STATA, so, differently to colleagues working in R and sharing data and codes, the Authors here must spend extra effort to convince the readers about their analysis. In short, full transparency is needed. The method section, however, is the weakest part of this paper. Analyses are explained poorly, and I suppose that most readers would struggle to understand and reproduce the same analysis:

As both reviewers had similar concerns we completely reworked our model selection/comparison approach. We rewrote the Methods/Data Analysis sections and reanalyzed the data using a simpler, more-robust strategy. See our responses to some of Reviewer 1 comments above on this issue.

- there is no mention of GLMM diagnostics, whether model assumptions are met

We added sentences in Data Analysis and Results of the GLMM diagnostics, model weights for comparisons. (Lines 190-194 and Lines 212-213)

- there is no mention of spatial and temporal autocorrelations, whether these have been accounted for, or whether significance patterns are affected by these issues

We added a Supplementary Information table of the correlations between variables for both Aim 1 analysis and Aim 2 analysis. None of the variables had correlations with a p-value of less than 0.05. For correlations between categorical variables we used Pearson's chi square and for mixed variables (a categorical variable and a continuous variable) we used logistic regression where the dependent variable was the categorical variable and independent variable was the continuous variable. Both Aim 1 or 2 had only one continuous variable each so none of the analyses required a correlation calculation between two continuous variables.

- in the method section, there is no mention of the multi-model comparison strategy (e.g., AIC or other similar methods) – they are used in the results, but not mentioned in the methods

We added sentences in the Methods to describe the metrics used in model comparison, weights, and change in AIC (Lines 190-194, and 212-213)

- in the result sections, there are statistical tests not mentioned in the methods (lines 257-262)

Added a sentence in Methods (Lines 190-194) to describe the approach to the descriptive statistics (z-tests to compare two population proportions), to match up to the Results section.

- the method section explaining alternative model structures is so confusing: why did you use a multi-model comparison in the first place? Why didn't you use a priori model structures considering your well-defined hypotheses? You need to justify your modelling approach to be credible to the readers

Thank for you for pointing this out. We changed our model selection approach - see above comments here and in Reviewer 1 responses.

- model structures seem a bit weak too, e.g. why year of study is not included as random intercept beyond the ID of the individuals!?

Year of sample was not included as a random intercept because we sampled wolves of all ages and did not know how long they had been infected. Many of them had likely been infected for years. *T. gondii* presents as a chronic infection, so individuals who have been infected at any point will test positive for the rest of their life. Controlling for year in this analysis would be unhelpful and perhaps unnecessarily confounding because it would lump together wolves captured in the same year even though they could have been infected recently or years prior. Individual ID was included as a random intercept because some wolves were captured and tested multiple times and we needed to account for repeatedly sampled individuals. For example, wolf 907F was captured at ages 0.7, 2.7, and 5.7. She tested negative at her first capture, and positive at her second and third. Some of her demographic variables changed over time (e.g., age, social status, cougar overlap index), and some did not change (e.g., coat color, sex). Including the ID random effect accounted for her repeated sampling and accounted for individual-level variation not captured in these variables.

- the method section mentions 4 metrics but lines 203-208 only refers to the modelling of dispersal behaviour – readers would guess whether you have forgotten to explain the other 3 models.

We changed this to clarify wording; it now reads “e.g., 1=dispersed or 0=did not disperse” (Line 224). Wolf dispersal is used as an example of how we coded a binomial distribution; this method was extended to the additional models where the behavior of interest was coded as a 1 and wolves that did not display that behavior were coded as a 0. The lines following the example of dispersal, 225-226, clarify the other behavior models used this same strategy.

To be fully appreciated, this paper needs a full rewriting of the method section, with a better clarification of sample sizes, modelling approaches, model assumptions, and more transparency on the results,

including output from STATA as supplementary material. The rest of the paper is simply great, this is a potentially outstanding story, but science needs robust empirical evidence first.

Thank you for your time and efforts and support of the overall goals of the paper. We have done a full rewriting of the Methods, Data Analysis, and Results to incorporate these comments and to improve transparency in our analysis approaches and results. Statistical output was reported in text (model weights) and so we were able to reduce the number of tables from the original submission (3 in the main document and 5 in the Supplemental Information) down to three in the main text and one in Supplemental Information. We edited the Introduction and some places in the Discussion as well to improve the background and transparency in our Data Analysis decisions, assumptions, and interpretations. Specifically, we expanded upon our decisions related to which variables to include in Aim 1 and Aim 2 and the scientific justification of some of the variables (cougar overlap index, time available, and system). Lastly, we made sure to clarify the difference between our results and subsequent interpretation and predictions with the discussion which included some interesting speculation for vital rate and ecosystem effects. We made sure these ideas are based on our work's results but included the appropriate qualifiers in the discussion.

We sincerely appreciate your time, effort, and support of our research's publication in *Communications Biology*. We hope that we have addressed any and all concerns you have brought up and had regarding this proposed publication. Thank you very much!

Sincerely,

Connor J. Meyer
Wildlife Biology Program
University of Montana
Missoula, MT 59812 USA
(360) 632-6015
connor.meyer@umontana.edu

Kira A. Cassidy
Yellowstone Wolf Project
Yellowstone Center for Resources
YNP, Wyoming, 82190 USA
(815) 353-6077
Kira_Cassidy@nps.gov

REVIEWERS' COMMENTS:

Reviewer #2 (Remarks to the Author):

This is an excellent story, really looking forward to seeing this in print. I have no more concerns and Authors have fully addressed all the points I have raised in the previous round of revisions.